# Increased Incidence of Thyroid Cancer among World Trade Center First Responders: A Descriptive Epidemiological Assessment

**DOI:** 10.3390/ijerph16071258

**Published:** 2019-04-09

**Authors:** Stephanie Tuminello, Maaike A. G. van Gerwen, Eric Genden, Michael Crane, Wil Lieberman-Cribbin, Emanuela Taioli

**Affiliations:** 1Institute for Translational Epidemiology and Department of Population Health Science and Policy, Icahn School of Medicine at Mount Sinai, New York, NY 10029, USA; Stephanie.Tuminello@mssm.edu (S.T.); maaike.vangerwen@icahn.mssm.edu (M.A.G.v.G.); wil.lieberman-cribbin@icahn.mssm.edu (W.L.-C.); 2Department of Otolaryngology-Head and Neck Surgery, Icahn School of Medicine at Mount Sinai, New York, NY 10029, USA; Eric.Genden@mountsinai.org; 3Division of Occupational and Environmental Medicine, Department of Environmental Medicine and Public Health, Icahn School of Medicine at Mount Sinai, New York, NY 10029, USA; michael.crane@mssm.edu; 4Department of Thoracic Surgery, Icahn School of Medicine at Mount Sinai, New York, NY 10029, USA; 5Tisch Cancer Institute, Icahn School of Medicine at Mount Sinai, New York, NY 10029, USA

**Keywords:** thyroid cancer, 9/11 disaster, World Trade Center, surveillance bias

## Abstract

An increased incidence of thyroid cancer among 9/11 rescue workers has been reported, the etiology of which remains unclear but which may, at least partly, be the result of the increased medical surveillance this group undergoes. This study aimed to investigate thyroid cancer in World Trade Center (WTC) responders by looking at the demographic data and questionnaire responses of thyroid cancer cases from the Mount Sinai WTC Health Program (WTCHP). WTCHP thyroid cancer tumors were of a similar size (*p* = 0.4), and were diagnosed at a similar age (*p* = 0.2) compared to a subset of thyroid cancer cases treated at Mount Sinai without WTC exposure. These results do not support the surveillance bias hypothesis, under which smaller tumors are expected to be diagnosed at earlier ages. WTCHP thyroid cancer cases also reported a past history of radiation exposure and a family history of thyroid conditions at lower rates than expected, with higher than expected rates of previous cancer diagnoses, family histories of other cancers, and high Body Mass Indexes (BMIs). Further research is needed to better understand the underlying risk factors that may play a role in the development of thyroid cancer in this group.

## 1. Introduction

An increased incidence of thyroid cancer has been reported in the Mount Sinai World Trade Center (WTC) responders [1], WTC-exposed fire fighters [2], and the New York City (NYC) Department of Health exposed residents [3] with an excess risk in the range of two to three times the incidence reported by New York City, New York State, and other national reference populations. This is despite the fact that established risk factors for thyroid cancer, such as exposure to radiation or iodine-131 [4,5], have not been reported in connection to Ground Zero [6]. The 9/11 attack did result in residents and responders being acutely exposed to several carcinogens mixed with dust and particles [1,7,8,9], although the direct link with WTC exposure and increased risk of cancer in WTC first responders remains unclear. Moreover, while the time and length of exposure of these responders have been reconstructed from questionnaires, this measure is clearly imprecise as recall bias is an important factor in a situation of extreme stress [10]. 

A possible explanation for the excess number of thyroid cancers observed in the responders cohort is over-diagnosis of thyroid cancer because of increased surveillance [11]. The large number of respiratory health issues developed by responders after their exposure to WTC dust has prompted a higher rate of diagnostic imaging of the chest, thus increasing the possibility of a chance thyroid nodule discovery [12,13]. Preliminary evidence, however, examining the thyroid nodules of WTC first responders found that all tissue samples tested were positive for biomarkers of malignancy, suggesting that while surveillance bias may be occurring, over-diagnosis of false-positives cannot sufficiently explain the excess risk of thyroid cancer observed in this group [14].

Disentangling the roles of various contributors to this excess risk of thyroid cancer would have major clinical and preventive consequences for the WTC responders. The aims of this study were to (a) examine non-modifiable demographic factors such as age, sex, race or ethnicity, as well as other contributing exposures predating thyroid cancer diagnosis; and (b) to better describe these WTC cancers and the similarities and differences of WTC thyroid cancers to non-WTC thyroid cancers.

## 2. Methods

### 2.1. Participant Selection and Enrollment

One of the cohorts in which excess thyroid cancer risk has been observed [1] is the Mount Sinai World Trade Center Health Program (WTCHP). Responders who participated (as employees or volunteers) in the rescue, recovery, and cleanup efforts at the WTC sites have been enrolled at Mount Sinai in the WTCHP, which is funded under the James Zadroga 9/11 Health and Compensation Act of 2010, on the basis of eligibility criteria including type of duties, site location, and dates and hours worked [15]. The medical protocol for the monitoring program includes self-administered physical and mental health questionnaires, as well as a physical examination, laboratory tests, spirometry, and a chest radiograph [15,16]. Over 27,000 responders have had a least one monitoring visit in the WTCHP and have consented for their data to be aggregated, of which a total of 20,984 have consented for their records to be used for medical research [15]. 

Cancer cases were identified through periodical linkage with the cancer registries of New York, New Jersey, Pennsylvania, and Connecticut, as these states account for 98% of the responders’ residencies at time of WTCHP enrollment. The full linkage methodology has been described elsewhere [15]. Only thyroid cancer cases validated by a cancer registry, and only those whose enrollment into the WTCHP pre-dated their cancer diagnosis, were considered eligible to participate in this study. Between 2002 and 2013 there were 73 eligible first responders who had developed thyroid cancer.

Eligible participants were contacted by phone, and those interested in participating were mailed an initial letter further explaining the research project, along with a consent form and brief questionnaire. This questionnaire was sectioned, with questions asking about past exposure to radiation, family medical history, personal medical history, smoking history, and other demographic information. Participants filled out the questionnaire at home and returned it by mail using a pre-stamped and pre-addressed return envelope. Participants had the contact information of the research assistant working on the project if they had any questions. In one case the questionnaire was filled out over the phone as the participant had trouble reading and writing. A signed copy of their study consents was mailed back to each participant. Of the 73 participants in the WTCHP cohort of responders who were eligible to be contacted, four had to be excluded because they did not speak English or because they had no viable contact information. Of the remaining 69 WTC thyroid cancer cases, 35 (51%) consented to completing the questionnaire. The study protocol was approved by the Icahn School of Medicine at Mount Sinai’s Institutional Review Board (IRB-17-01323).

### 2.2. Data Processing and Statistical Analysis

Upon return of the questionnaires, the data were input into excel to create a database of questionnaire responses, and this database was uploaded into SAS for statistical analysis. Some questionnaire responses were collapsed to make results more interpretable. If a participant reported having had an X-ray of their head or neck, a computed tomography (CT) scan of their head, neck or chest, or having had radiation therapy in the head or neck region predating their thyroid cancer diagnosis they were considered as having had some sort of radiation exposure. A diagnosis of another cancer was recorded and specified if that cancer diagnosis predated or postdated their diagnosis of thyroid cancer. BMI was calculated using the self-reported height and weight of the study participants, and with the cutoffs used for underweight, overweight and obese were <25, 25–30 and >30 kg/m^2^, respectively, as suggested by the Center for Disease Control and Prevention [17]. 

The data obtained from the questionnaires, as well as the initial data reported by the WTCHP (age at diagnosis, race, gender, histology, marital status, education and smoking status) were compared to a sample of thyroid cancer from the Mount Sinai Cancer Registry generated by an official from the Cancer Registry. This sample was limited to those diagnosed with thyroid cancer and treated at Mount Sinai between 2002 and 2013, in order to cover a comparable time-frame. 

Chi-square test, or Fisher’s exact test for small sample size were used to compare WTC and Sinai Registry data for categorical variables. Wilcoxon Rank Sum Test was used for continuous variables not normally distributed. All statistical analysis was conducted using SAS version 9.4 (SAS Institute Inc., Cary, NC, USA). 

## 3. Results

There were 73 thyroid cancer cases from the WTCHP cohort and these were compared to 949 thyroid cancer cases from the Mount Sinai Cancer Registry. The majority of thyroid cancer cases were white, both in the WTCHP group (71.9%) and in the Mount Sinai Cancer Registry (70.2%) (*p* = 0.8). There was, however, a statistically significant difference in gender (*p* < 0.0001), with WTCHP cases more likely to be male (78.1%). There was no significant difference between the two groups in terms of tumor size (*p* = 0.4), and the mean tumor size was small in both the WTCHP and the Mount Sinai group, 1.4 cm and 1.8 cm, respectively. Age at diagnosis was also similar between the two groups (*p* = 0.2); the mean age for those diagnosed after WTC exposure was 48.9 years while the Mount Sinai Registry mean diagnosis age was 51 years old. There was a statistically significant difference in terms of histology (*p* = 0.04), with those in the WTCHP cohort more likely to be diagnosed with a subtype of papillary carcinoma. There was also a statistically significant difference between the WTCHP and Mount Sinai group in terms of smoking (*p* = 0.0385), whereby those in the WTCHP group were more likely to be current smokers, and in terms of marital status (*p* < 0.0001), with those in the WTCHP cohort being more likely to be married (Table 1).

Out of the 69 eligible WTC thyroid cancer cases, 35 (51%) consented to complete the questionnaire. Those who consented and completed the questionnaire did not differ significantly in terms of race, gender, age at diagnosis or histology than those who did not participate in the study (data not shown). A total of 23% of cases reported some sort of radiation exposure. While only 9% reported a family history of thyroid health issues (either a benign goiter or thyroid cancer), 67% reported a family history of other cancer types. Additionally, 21% reported a previous history of cancer before the diagnosis of thyroid cancer. The majority of the study cohort had a BMI between 25 and 30 Kg/m^2^ (44%) or >30 kg/m^2^ (44%). Most participants (62%) reported that their thyroid cancer was diagnosed as a consequence of routine or incidental medical surveillance, and not because they went to a doctor with symptoms consistent with thyroid cancer (Table 2). 

## 4. Discussion

Whether or not surveillance bias is occurring in the WTCHP cohort, and the extent to which this bias may be contributing to the increased incidence of thyroid cancer among first responders, remains unclear. The similar clinical characteristics observed between the WTCHP responders and those in the Mount Sinai Registry suggest that the excess risk of thyroid cancer in first responders cannot be adequately explained by surveillance bias alone. Under the surveillance bias hypothesis, we would expect increased detection of small thyroid nodules [11,18], yet there was no statistical difference between the average tumor size of the WTCHP and Mount Sinai Registry groups. Moreover, the average age at diagnosis was similar between the two groups, while if surveillance bias was introduced then smaller cancers should have been detected at an earlier age in the WTC cohort. It is important to consider how surveillance bias may be occurring across the US as a whole; previous research has found that much of the national increase in thyroid cancer incidence can be attributed to an increase in small (<1 cm) tumors, unlikely to be found in the absence of routine or incidental surveillance [11]. Moreover, the majority of WTC participants reported that their cancers were diagnosed due to routine or incidental screening, which may be higher than the expected rate of asymptomatic thyroid cancer detection [19].

The WTCHP and Mount Sinai thyroid cancer groups statistically differed in terms of gender, histology, smoking history and marital status; they were not statistically different in terms of race. Comparison of gender between the two groups, however, is difficult to interpret, since the WTCHP is composed predominantly of males [20]. WTCHP thyroid cancer cases were more likely to be papillary histology, which tend to have a favorable prognosis [21]. Moreover, those in the WTCHP cohort were more likely to be current smokers when compared to the Mount Sinai Registry group, though the number of reported current smokers was small for both groups. It remains controversial what affect smoking has on thyroid cancer risk; a decrease in risk of thyroid cancer in men who are current smokers has actually been observed, although there is no known biological justification for this [22,23]. WTCHP respondents with thyroid cancer were also more likely to be married or have a partner; data indicate reduced mortality from cancer associated with being married as opposed to being single, possibly because of increased social support [24]. 

While WTCHP thyroid cancer cases do not appear to be clinically distinct, their reported risk factors and carcinogenic exposure history may be uncommon. Only about 23% reported having had some sort of previous diagnostic radiation exposure before their thyroid cancer diagnosis, while other studies have found that this number could be as high as 85%, even suggesting this type of radiation exposure contributes to the carcinogenic process [25].

Increased BMI is also associated with increased risk of thyroid cancer [26] and 44.1% of WTCHP responders reported a BMI 25–29 km/m^2^, with an additional 44.1% reporting a BMI >30 km/m^2^. Neta et al. reported 15.9% and 12% of the thyroid cancer cases having a BMI of 25–29, and ≥30 km/m^2^, respectively, which is lower than what was observed in the WTCHP cohort. However, these percentages seem to be in keeping with what has been reported for the overall WTCHP general responder cohort, suggesting that the thyroid cancer cases represent a random subset of the total group in terms of BMI [25]. 

Moreover, few WTCHP responders (8.8%) reported in their questionnaire that they had a family history of thyroid cancer or benign thyroid issues. Having a first-degree relative with thyroid cancer is known to increase thyroid cancer risk [27]. Other studies have found that as high as 15.6% of thyroid cancer cases report a family history of thyroid cancer [25,28]. A family history of another malignant disease has also been shown to be associated with thyroid cancer [29], and having a family history of cancer was reported by 66.7% of WTCHP respondents. This is higher than reported by past research, which found that 49% of thyroid cancer cases reported a family history of cancer in first-degree relative [29]. It is possible that some WTC respondents may have reported past cancers of relatives more distant than first-degree relatives, thus inflating the statistic. 

Having a previous cancer diagnosis has also been shown to be associated with thyroid cancer [30], and 20.6% of WTCHP respondents reported having had another cancer before being diagnosed with thyroid cancer. This is higher than what was reported in the SEER database [31], where only 10.9% of thyroid cancer cases had another cancer predating thyroid cancer diagnosis. It is possible that this is related to the fact that WTC responders are at an increased risk for several types of cancer, including prostate cancer [1]; an alternative explanation could be an increased familial risk among WTCHP responders. 

The WTCHP responders were also likely (23.5%) to develop a second primary cancer after thyroid cancer, which is in keeping with the observed increase in risk of secondary cancers associated with a primary thyroid cancer [32]. This number is higher, however, than what would be expected based on SEER registry data, whereby just 8% of thyroid cancer cases developed a second type of tumor. Again, it remains unclear if this increased risk of multiple primary cancers is because of WTC dust and debris exposure having a carcinogenic effect on 9/11 first responders or because of other genetic or environmental factors. 

This study had some limitations, which includes a small sample size that may have been affected by recall bias, since participants were asked about past exposure and family history. Although selection bias could have been possible, an attempt was made to verify that those who participated were not statistically different from those who chose not to. 

Among the strengths, this study represents an important contribution to the literature by helping to fill the gap in understanding why 9/11 first responders experience an increased risk of thyroid cancer. The descriptive epidemiology presented here will hopefully inform future, more in-depth studies that may explain this phenomenon. Future research on germline and somatic tumor alterations of these cancers may help to shed light on the possibility of a WTC-related carcinogenic mechanism. 

## 5. Conclusions

Cancer surveillance of WTC first responders should continue, and specifically thyroid health should be part of regular screening procedures. Ultrasound techniques instead of radiation-based diagnostic procedures might be a more appropriate first step approach.

## Figures and Tables

**Table 1 ijerph-16-01258-t001:** Description of World Trade Center Health Program (WTCHP) and Mount Sinai Registry thyroid cancer cases.

Demographic and Clinical Characteristics *	WTCHP (*n* = 73)		Sinai Cancer Registry (*n* = 949)		*p* value
	*n*	%	*n*	%	
Race/Ethnicity					0.8177
White	41	71.9	634	70.2	
Black	3	5.3	68	7.5	
Other	13	22.8	201	22.3	
Gender					<0.0001
Female	16	21.9	695	73.3	
Male	57	78.1	253	26.7	
Tumor Size (cm)	Mean (1.4)	SD (1.2)	Mean (1.8)	SD (1.9)	0.4053
Age at Diagnosis	Mean (48.9)	SD (8.0)	Mean (51.0)	SD (16.2)	0.2252
Histology ^					0.0363
*Papillary Carcinoma*	66	90.4	792	83.5	
*Papillary Adenocarcinoma NOS*	(63.6%)		(58.7%)		
*Papillary Carcinoma Follicular Variant*	(28.8%)		(22.0%)		
*Papillary Microcarcinoma*	(3.0%)		(17.7%)		
*Papillary Carcinoma Columnar Cell*	(4.5%)		(1.6%)		
Other Adenocarcinoma	5	6.9	41	4.3	
*Oxyphilic Adenocarcinoma*	(60.0%)		(46.3%)		
*Follicular Adenocarcinoma*	(40.0%)		(53.7%)		
Other	2	2.7	116	12.2	
Smoking Status					0.0385
Current Smoker	3	5.1	14	1.5	
Former or Never Smoker	56	94.9	925	98.5	
Marital Status					<0.0001
Married or Partnered	42	70.0	519	59.7	
Separated or Divorced	12	20.0	56	6.4	
Single	3	5.0	241	27.7	
Widowed	3	5.0	53	6.1	

* Race *n*_WTC_ = 57, *n*_registry_ = 903; gender *n*_WTC_ = 73, *n*_registry_ = 948; tumor size *n*_WTC_ = 27, *n*_registry_ = 753; age at diagnosis *n*_WTC_ = 73, *n*_registry_ = 949; histology *n*_WTC_ = 73, *n*_registry_ = 94, marital status *n*_WTC_ = 60, *n*_registry_ = 869. ^ Histology is based on International Classification of Diseases for Oncology (ICD-O-3) coding: Papillary adenocarcinoma NOS = 8260, oxyphilic adenocarcinoma = 8290, follicular adenocarcinoma NOS = 8330, papillary carcinoma follicular variant = 8340, papillary microcarcinoma = 8341, papillary carcinoma columnar cell = 8344.

**Table 2 ijerph-16-01258-t002:** WTC thyroid cases questionnaire responses (*n* = 35).

Exposures and Medical History *	*n*	%
Radiation Exposure		
Yes	8	22.9
No	27	77.1
Family History of Thyroid Issues		
Yes	3	8.8
No	31	91.2
Family History Other Cancer		
Yes	22	66.7
No	11	33.3
Personal History of Another Cancer		
Before Thyroid Cancer	7	20.6
After Thyroid Cancer or Unknown dx Date	8	23.5
No Other Cancer	19	55.9
BMI (kg/m^2^)		
<25	4	11.8
25–30	15	44.1
>30	15	44.1
Diagnosis Method		
Because of Symptoms	21	61.8
Due to Routine Screening or Unrelated Medical Event	13	38.2

* Missing data for family history of other cancer (1), family history of other cancer (2), personal history of other cancer (1), smoking history (1), BMI (1), diagnosis method (1).

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
