# Peer review of "Increased Incidence of Thyroid Cancer among World Trade Center First Responders: A Descriptive Epidemiological Assessment"

_ijerph, 2019, doi:10.3390/ijerph16071258_

Round 1

Reviewer 1 Report

In the decade plus following the World Trade Center disaster, first responders and residents living near the WTC experienced a 2-3 fold increased in diagnosis of thyroid cancer. The current paper addressed one of the possible artifacts that might have contributed to the reported high diagnostic rate: that increased medical surveillance of the WTC exposed cohort would have led to higher incidence of thyroid nodules and smaller cancers.  The authors also compared demographic factors of participants from the WTC cohort and compared those to large numbers of thyroid cancer patients, diagnosed during the same years and registered at the same facility in NYC.

The lack of difference in tumor size and age at diagnosis did not support the hypothesis that the increased rate of thyroid cancer in the WTC cohort was the result of surveillance bias.

A subset of the WTC cohort also completed questionnaires addressing previous radiation exposure and thyroid and other cancers before and after diagnosis with thyroid cancer. Members of the WTC cohort reported lower levels of previous radiation exposure and lower incidences of familial thyroid cancers, both risk factors for developing thyroid cancer. On the other hand, the members of the WTC cohort reported an increased familial history of other cancers, and increased history of their own other cancers, both before and after their diagnosis with thyroid cancer.

The authors have presented the results and possible implications of these data in a clear, thoughtful manner.  The article is of importance in directly addressing one critique of interpretations of WTC negative health outcomes, that of potential surveillance bias.  The data that indicate lower levels of two known risk factors for the disease support the need to further study the etiology of thyroid cancer in this important cohort.  The final suggestion, based on the increased observation of all cancer in the WTC cohort, is that enhanced immunological competence and epigenetic changes in gen regulation caused be the result of WTC-related exposures.

I believe this paper can be published, as is.

Author Response

Response: We thank reviewer 1 for taking the time to review our work; we appreciate that they feel it is of sufficient quality to be published in the International Journal of Environmental Research and Public Health. 

Reviewer 2 Report

The manuscript ‘Risk factors for thyroid cancer in World Trade Center first responders’ by Tuminello et al performed statistical analysis by comparing the patient characteristics (race, gender, tumor size, age at diagnosis, histology, smoking status, marital status) of thyroid cancer patients who has or not has 911 exposures. Significant differences were found in gender, histology and marital status, but not in tumor size or age at diagnosis, suggesting that 911 exposure might play a role in increased risk of thyroid cancers. 

The manuscript is well organized. The data are mainly descriptive. 

Author Response

Response: We agree with reviewer 2 that our work here is mainly descriptive. Thus we have addressed any language that might be too strong, and have edited our title to be better reflective of the descriptive nature of our work. The new title is “Increased incidence of thyroid cancer among World Trade Center first responders: a descriptive epidemiological assessment”

Reviewer 3 Report

This manuscript presents a descriptive study examining thyroid cancer among the World Trade Center Health Program participants.  The authors provide descriptive demographic and clinical information for the 73 thyroid cancer cases diagnosed in this cohort, compared to a selection of thyroid cancer cases from the Mount Sinai Cancer Registry.  Based on similarities between the two groups, they rule out the surveillance bias hypothesis to explain potentially higher incidence of thyroid cancer incidence among the WTCHP group. They then present some self-reported information collected from a subset of WTCHP thyroid cancer cases on radiation exposure history, personal and family history of cancer, and BMI.  They conclude from these data that WTC exposure may put this group at higher risk of thyroid cancer. 

My primary concern with this paper is that I do not believe the study design suitable to examine risk. Therefore, I believe that the data presented cannot substantiate the conclusions drawn by the authors.  While conclusions should always be well supported by the data, it is my belief that not doing so in this case could be particularly irresponsible, since this topic may be of public interest, given the group involved. I have described below specific sections in which the authors need to scale back their language.  This is a descriptive study, and should be presented as such, and only as such.

Title: Please remove mention of risk factors.  Suggest something like “Descriptive epidemiology of thyroid cancers among WTC FR.”

Abstract: Please remove conclusion statement regarding increased risk of cancer among the WTC FR group.

Line 47, introduction.  Statement that “excess risk 2-3 times the incidence reported by cancer registries” is confusing to me.  Not least because these individuals will themselves be recorded in the registry.  Do the authors mean compared to the general US population?

Line 58, introduction. What do the authors mean by “true malignancies”?  How are they defining this?

Lines 62-64, introduction: These aims are not in line with the abstract and the conclusions.  They are more along the lines of the “descriptive epi” that I believe can be supported by the data presented, rather than the “risk factor exploration” that is presented elsewhere.

Methods: Please describe the survey return rate in the methods, not in the results.

Methods: In the second paragraph, please list ICD-O codes included.  Please describe histology codes used to classify thyroid cancers.  Please confirm whether classification includes the recent proposal for the downgrade of NIFTP (Jama oncology, 2016)

Methods: No information is given on the selection of the Sinai cases used for comparison.  This information needs to be included. It is very difficult to assess the appropriateness of this comparison group (and in turn the validity/quality of this work) without this information.

Methods: The WTCHP needs to be described.  Is this a healthcare program? A research study?

Methods, line 100: If a cancer diagnosis post-dated the diagnosis of thyroid cancer, then it is NOT a previous history of cancer.  Please change this definition, as it could impact results.

Methods: The authors mention a comparison to SEER (which database?), but I cannot see this presented in the results?

Results: Table 1 should also present demographic information for the complete WTCHP cohort, for relevant variables.

Results (should be methods): the response rate for the survey is low, begging the question of selection bias. This issue is not mentioned by the authors. Were the demographic characteristics of those who responded similar to those who did not? This should be presented in the methods so that the reader can assess whether or not the data are missing at random.

Results should describe the findings of the table, not just provide a list of the p-values from the tables.

Table 2: These findings provide information on these potential risk factors for thyroid cancer in this group, but they CANNOT provide any information about whether this group is at increased risk.  At the very least, there is no comparison group.

Discussion, lines 209 – 212 and elsewhere. Please remove all language that suggests that WTC FR may be at increased risk of thyroid cancer.  The data cannot support this.

Discussion: This section also contains a lot of speculative language around the potential for WTC events causing cancer, in a way that implies that the authors support this belief. They do not adequately discuss other reasons this group may differ in thyroid cancer incidence. I would like to see this language scaled back to that which can be reasonably supported by the science.  Or at the very least provide a more balanced view.  I have concerns that this discussion might cause (as yet) undue concern for this population.

Discussion: This section is very long (and needs substantial shortening), but is missing some critical information. For example, a discussion of the potential for recall bias from the questionnaire would make any comparison of these data to similar data questionable.  Furthermore, the potential for selection bias in the questionnaire responses.

Author Response

Comments and Suggestions for Authors

This manuscript presents a descriptive study examining thyroid cancer among the World Trade Center Health Program participants.  The authors provide descriptive demographic and clinical information for the 73 thyroid cancer cases diagnosed in this cohort, compared to a selection of thyroid cancer cases from the Mount Sinai Cancer Registry.  Based on similarities between the two groups, they rule out the surveillance bias hypothesis to explain potentially higher incidence of thyroid cancer incidence among the WTCHP group. They then present some self-reported information collected from a subset of WTCHP thyroid cancer cases on radiation exposure history, personal and family history of cancer, and BMI.  They conclude from these data that WTC exposure may put this group at higher risk of thyroid cancer. 

My primary concern with this paper is that I do not believe the study design suitable to examine risk. Therefore, I believe that the data presented cannot substantiate the conclusions drawn by the authors.  While conclusions should always be well supported by the data, it is my belief that not doing so in this case could be particularly irresponsible, since this topic may be of public interest, given the group involved. I have described below specific sections in which the authors need to scale back their language.  This is a descriptive study, and should be presented as such, and only as such.

Title: Please remove mention of risk factors.  Suggest something like “Descriptive epidemiology of thyroid cancers among WTC FR.”

Response: The title has been changed from “Risk factors for thyroid cancer in World Trade Center first responders “ to “Increased incidence of thyroid cancer among World Trade Center first responders” a descriptive epidemiological assessment” in an effort to incorporate the language suggested by reviewer 3.

Abstract: Please remove conclusion statement regarding increased risk of cancer among the WTC FR group.

The sentences “These results suggest that WTC responders may be at increased risk of multiple cancers, including thyroid cancer, where WTC exposure may have a role. Further studies on the biology of these tumors are warranted” has been removed from the abstract as suggested, and instead replaced with “Further research is needed to better understand the underlying risk factors that may play a role in thyroid cancer development in this group.”

Line 47, introduction.  Statement that “excess risk 2-3 times the incidence reported by cancer registries” is confusing to me.  Not least because these individuals will themselves be recorded in the registry.  Do the authors mean compared to the general US population?

Response: Different studies used different reference populations, although each found an increased risk of thyroid cancer in the range reported in our introduction (2-3 times as high). Solan et al. used New York State Cancer Registry data as comparison, Zeig-Owens et al. used US National Cancer Institute Surveillance Epidemiology and End Results (SEER), and Jordan used New York City rates to derived expected incidence of thyroid cancer. To clarify, the first sentence in the introduction has been updated to read “An increased incidence of thyroid cancer has been reported in the Mount Sinai World Trade Center (WTC) responders,[1] WTC-exposed fire fighters,[2] and the NYC Department of Health exposed residents,[3] with an excess risk in the range of 2 to 3 times the incidence as reported by New York City, New York State, and other national reference populations.”

Line 58, introduction. What do the authors mean by “true malignancies”?  How are they defining this?

Response: Here we mean to say that the thyroid tissue samples from WTC responders have been molecularly confirmed to be cancer tissue, as thus are considered to be “true malignancies” instead of false-positive diagnoses. To clarify, we have edited the last line of paragraph two of the introduction to read “Preliminary evidence, however, examining the thyroid nodules of WTC first responders found that all tissue samples tested were positive for biomarkers of malignancy, suggesting that while surveillance bias may be occurring, over-diagnosis of false-positives cannot sufficiently explain the excess risk of thyroid cancer observed in this group.”  

Lines 62-64, introduction: These aims are not in line with the abstract and the conclusions.  They are more along the lines of the “descriptive epi” that I believe can be supported by the data presented, rather than the “risk factor exploration” that is presented elsewhere.

Response: We agree with the reviewer that the language used in our manuscript should be altered to better reflect the descriptive nature of our analysis. Towards that end we have kept the study aims as written in the introduction, as suggested by the reviewer, and instead updated the language used in the abstract when discussing the aims of our project. The abstract now reads “An increased incidence of thyroid cancer among 911 rescue workers has been reported, the etiology of which remains unclear but which may, at least partly, be the result of the increased medical surveillance this group undergoes. This study aimed to investigate thyroid cancer in WTC responders by looking at the demographic data and questionnaire responses of the Mount Sinai WTC Health Program (WTCHP) thyroid cancer cases.”

Methods: Please describe the survey return rate in the methods, not in the results.

Response: we agree with reviewer 3 that survey participation is an important goal to be considered when planning study design, and thus we moved the section on return rates to the methods; we also briefly mention the return rates in the results as this provides important context for comprehension of the rest of the results.

Methods: In the second paragraph, please list ICD-O codes included.  Please describe histology codes used to classify thyroid cancers.  Please confirm whether classification includes the recent proposal for the downgrade of NIFTP (Jama oncology, 2016)

Response: All cancers examined fell under the ICD-O-3 topography code C73, although eligibility was not limited by further histological subtype. The specific ICD-O-3 codes used to classify histologies have been listed under table 1 as per reviewer 3’s suggestion. Unfortunately, all our WTC cases were diagnosed before 2016 and we don’t have pathology reports for all of the WTC cases. We have worked with de-identified Mount Sinai data, and thus we cannot go back and recode the Papillary Carcinoma Follicular Variant cases (nwtc=19).  Although it would be important to know if any of these 19 cases could be downgraded, we are not studying outcomes in this manuscript.

Methods: No information is given on the selection of the Sinai cases used for comparison.  This information needs to be included. It is very difficult to assess the appropriateness of this comparison group (and in turn the validity/quality of this work) without this information.

Response: The selection of the Mount Sinai cases used for comparison is specified in the methods section, lines 117-120, and reads “were compared to a sample of thyroid cancer from the Mount Sinai Cancer Registry generated by an official from the Cancer Registry and was limited to those diagnosed with thyroid cancer and treated at Mount Sinai between the years of 2002 and 2013, in order to cover a comparable time-frame.” Further selection limits were not used as this was meant to be a random subset of thyroid cancer patients.

Methods: The WTCHP needs to be described.  Is this a healthcare program? A research study?

Response: The WTCHP is a monitoring program based at Mount Sinai. To clarify this the authors have included additional information about the program in the first paragraph, second sentence of the methods section, “The medical protocol for the monitoring program includes self-administered physical and mental health questionnaires, as well as a physical examination, laboratory tests, spirometry, and a chest radiograph.”

Methods, line 100: If a cancer diagnosis post-dated the diagnosis of thyroid cancer, then it is NOT a previous history of cancer.  Please change this definition, as it could impact results.

Response: table 2 reports details on whether or not participants had an additional cancer diagnosis, and if that cancer diagnosis predated or postdated their thyroid cancer diagnosis. We agree that the wording as written in the methods section was misleading, and have changed it to read “A diagnosis of another cancer was recorded and specified if that cancer diagnosis predated or postdated the thyroid cancer diagnosis.”

Methods: The authors mention a comparison to SEER (which database?), but I cannot see this presented in the results?

Response: SEER (Surveillance, Epidemiology and End Results) data is publically available, and was used to look at cancer history in those with thyroid cancer. As this data was not used for any statistical analysis, and only mentioned as a reference in the discussion, this is not an included part of our results. The SEER database is cited, and for clarity we have forgone discussing SEER in relation to our methods.   

Results: Table 1 should also present demographic information for the complete WTCHP cohort, for relevant variables.

Response: Table 1 does present demographic for the complete WTCHP thyroid cancer cohort (n=73) from which we were recruiting participants. Unfortunately we do not have detailed demographic data on the entire WTC cohort to date.

Results (should be methods): the response rate for the survey is low, begging the question of selection bias. This issue is not mentioned by the authors. Were the demographic characteristics of those who responded similar to those who did not? This should be presented in the methods so that the reader can assess whether or not the data are missing at random.

Response: We have added a sentence to the second paragraph of the results section to illustrate that those who participated in our study were not statistically significantly different in demographic or clinical characteristics from those who choose not to participate. It reads “Those who consented and completed the questionnaire did not differ significantly in terms of race, gender, age at diagnosis or histology than those who did not participate in the study (data not shown).”

Results should describe the findings of the table, not just provide a list of the p-values from the tables.

Response: The results section has been improved and expanded upon as suggested.

Table 2: These findings provide information on these potential risk factors for thyroid cancer in this group, but they CANNOT provide any information about whether this group is at increased risk.  At the very least, there is no comparison group.

Response: we agree. We have removed the term “Risk” from the title of Table 2.

Discussion, lines 209 – 212 and elsewhere. Please remove all language that suggests that WTC FR may be at increased risk of thyroid cancer.  The data cannot support this.

Response: We agree, and the language throughout the discussion has been changed to be more cautious and appropriate.

Discussion: This section also contains a lot of speculative language around the potential for WTC events causing cancer, in a way that implies that the authors support this belief. They do not adequately discuss other reasons this group may differ in thyroid cancer incidence. I would like to see this language scaled back to that which can be reasonably supported by the science.  Or at the very least provide a more balanced view.  I have concerns that this discussion might cause (as yet) undue concern for this population.

Response: Agreed, the last paragraph of the discussion has been removed in an effort to keep our discussion within the bounds of what our results support.

Discussion: This section is very long (and needs substantial shortening), but is missing some critical information. For example, a discussion of the potential for recall bias from the questionnaire would make any comparison of these data to similar data questionable.  Furthermore, the potential for selection bias in the questionnaire responses.

Response: As suggested, the discussion section has been shortened. We have also attempted to more fully elaborate on the potential limitations that review 3 points out in the last paragraph “This study has some limitations: it includes a small sample size, and may be affected by recall bias, since participants were asked about past exposure and family history; although selection bias could have been possible, we made an attempt to verify that those who participated were not statistically different from those who did not. Among the strengths, our study represents an important contribution to the literature by helping to fill the gap in understanding why 911 first responders experience an increased risk for thyroid cancer. The descriptive epidemiology presented here will hopefully inform future, more in-depth studies that may explain this phenomenon. Future research on germline and somatic tumor alterations of these cancers may help shed light on the possibility of a WTC-related carcinogenic mechanism.”

Reviewer 4 Report

Dear Authors

In my opinion the theme of the article is very actual and interesting for the readers of the journal.

The manuscript under revision aimed to investigate risk factors of thyroid cancer based on demographic data and questionnaire responses of the Mount Sinai WTC Health Program thyroid cancer cases.

The authors found that WTCHP thyroid cancer tumors were of a similar size and were diagnosed at a similar age compared to a subset of thyroid cancer cases treated at Mount Sinai without 911 exposures.

WTCHP thyroid cancer cases also reported a past history of radiation exposure and a family history of thyroid conditions at lower rates than expected, and higher than expected rates of previous cancer diagnoses, family histories of other cancers, and high BMIs.

These authors’ results suggest that WTC responders may be at increased risk of multiple cancers, including thyroid cancer, where WTC exposure may have a role. Cancer surveillance of WTC first responders should therefore continue, and specifically thyroid health should be part of regular screening procedures; ultrasound techniques instead of radiation- based diagnostic procedures might be a more appropriate first step approach

In my opinion only minor revision is needed, please see attached file.

Best regards

Author Response

Response: The file indicated contained only the original manuscript. We communicated this to our editor Dr. Robert Zhao, who indicated that it would be ok to proceed with the revisions as suggested by the other reviewers. We remain happy to consider any changes suggested by reviewer 4 in future.